# Ion-cluster-mediated ultrafast self-healable ionoconductors for reconfigurable electronics

Yong Min Kim [1], Jin Han Kwon[1], Seonho Kim [2], U Hyeok Choi[2] & Hong Chul Moon [1✉]

Implementing self-healing capabilities in a deformable platform is one of the critical challenges for achieving future wearable electronics with high durability and reliability. Conventional systems are mostly based on polymeric materials, so their self-healing usually proceeds at elevated temperatures to promote chain flexibility and reduce healing time. Here, we propose an ion-cluster-driven self-healable ionoconductor composed of rationally designed copolymers and ionic liquids. After complete cleavage, the ionoconductor can be repaired with high efficiency (∼90.3%) within 1 min even at 25 °C, which is mainly attributed to the dynamic formation of ion clusters between the charged moieties in copolymers and ionic liquids. By taking advantages of the superior self-healing performance, stretchability (∼1130%), non-volatility (over 6 months), and ability to be easily shaped as desired through cutting and re-assembly protocol, reconfigurable, deformable light-emitting electroluminescent displays are successfully demonstrated as promising electronic platforms for future applications.

[1] Department of Chemical Engineering, University of Seoul, Seoul 02504, Republic of Korea. [2] Department of Polymer Science and Engineering and Program in Environmental and Polymer Engineering, Inha University, Incheon 22212, Republic of Korea. ✉email: hcmoon@uos.ac.kr

Living creatures in nature can automatically heal themselves when injured, which is an attractive function to extend their lifespans. However, synthetic materials lack the ability to withstand repeated mechanical stress and damage, which is a technical obstacle to the development of sustainable platforms. Therefore, the concept of self-healing has been proposed and extensively studied for various applications, including highly durable soft actuators[1-3], optoelectronics[4,5], field-effect transistors[6,7], and energy devices[8-10]. Furthermore, self-healing materials are expected to maximize the utility of electronic equipment in extreme conditions not easily accessible to humans, such as space and deep-sea environments.

One representative strategy to implement self-healing capabilities is incorporating healing agent-containing microcapsules into a matrix, which is referred to as the extrinsic method[11,12]. As the microcapsules break, the healing agent (e.g., epoxy resins and liquid metals) fills the cracks when damaged. However, because the agent is consumed with no regeneration, the resulting systems cannot undergo multiple self-healable cycles. To overcome this limitation, a reversible bond-mediated protocol was suggested as a way to obtain sustainable self-healing materials. Disulfide/diselenide/ditelluride bonds[13-15], transesterification[16], and Diels-Alder reactions[17,18] are possible examples of reversible covalent bonds used in the healing process. However, most covalent candidates require external energy sources, such as heat and UV, to promote bond cleavage-formation reaction kinetics[14-18]. The use of relatively weak non-covalent bonds (e.g., hydrogen bonding[19,20], hydrophobic interaction[21], and electrostatic interaction[22-25]) allows the damage recovery at low temperatures, but the poor mechanical robustness (typically, Young's modulus <100 kPa) of non-covalent bond-driven self-healing materials limits their applications. The combination of covalent and non-covalent bonds in a single system was proposed as a possible alternative, whereas healing at elevated temperatures is still required to facilitate polymeric chain mobility and rapid self-healing[14,26,27]. Therefore, developing breakthrough self-healing mechanisms to simultaneously achieve mechanical robustness and rapid restoration at low temperatures is still a critical challenge.

Herein, we propose an innovative self-healing process driven by the rapid dynamic formation/dissociation of ion clusters (ICs) to overcome the conventional trade-off between mechanical properties and healing performance. To this end, we designed ionoconductors consisting of a charged-moieties-containing copolymer matrix, poly([(3-acryloamidopropyl)-trimethylammonium][bis(trifluoromethanesulfonyl)imide])-ran-poly(2-hydroxyethyl acrylate) (P[AA][TFSI]-r-PHEA), doped with a room-temperature ionic liquid, 1-ethyl-3-methylimidazolium bis(trifluoromethylsulfonyl)imide ([EMI][TFSI]). In contrast to previous self-healing systems where the interaction between polymer chains is important, the charged domains of the copolymers exploited the added [EMI][TFSI] as a mediator to form ICs. This resulted in extremely rapid (sub-1 min) self-healing even at low temperatures (e.g., 25 °C), because no significant movement of polymer chains was needed. The self-healing performance of the ionoconductor in this work corresponds to the top-tier level among reported systems in terms of two critical parameters: healing temperature and time. Furthermore, the generated ICs served as physical crosslinking points, leading to a high mechanical robustness of the ionoconductors. More interestingly, we successfully exploited the unprecedentedly fast self-healing characteristics of the present system to demonstrate new deformable electronics platforms, such as area-adjustable iono-skin sensors and user-reconfigurable AC electroluminescent displays (ACEDs). Overall, the results show a breakthrough IC-driven strategy and unveil the origin of the ultra-fast self-healing behavior, which opens up a new opportunity for future device platforms of reconfigurable electronics.

## Results

**Rational design and physical properties of ultrafast self-healable ionoconductors.** To overcome the trade-off between high self-healing performance and mechanical robustness of conventional systems[2,3,14,19-38] (Fig. 1a), we designed a P[AA][TFSI]-r-PHEA (AAHA) copolymer containing charged pendant groups (P[AA][TFSI]) and hydrogen bonding sites (PHEA), and prepared it by a simple one-pot copolymerization method (Supplementary Fig. 1). The introduction of [EMI][TFSI] generated non-covalently bound ICs through electrostatic interaction with P[AA][TFSI] in the copolymer, leading to IC-mediated supramolecular networks (Supplementary Fig. 2). To elucidate the role of the IC formation, we also synthesized a non-ionic copolymer, poly(N-[3-(dimethylamino)propyl]acrylamide)-ran-poly(2-hydroxyethyl acrylate) (PDA-r-PHEA, DAHA). The molecular characteristics of all copolymers prepared in this work are summarized in Supplementary Table 1. For fairness, we compared ionoconductors based on 92-AAHA and 90-DAHA, denoted as 92-AAHA-IL and 90-DAHA-IL, respectively, because both were based on copolymers with similar PHEA fractions (Supplementary Fig. 3) and total molecular weights (Supplementary Fig. 4). We note that the numbers in the sample codes indicate the mole fractions of P[AA][TFSI] and PDA in the copolymers. The content of [EMI][TFSI] in the two ionoconductors was fixed at 30 wt%.

Figure 1b displays the master curves of 92-AAHA-IL and 90-DAHA-IL at a reference temperature of 20 °C, in which the time–temperature superposition (tTS) principle was applied to the dynamic storage ($G'$) and loss ($G''$) moduli profiles at various temperatures (Supplementary Fig. 5). Depending on the presence of charged moieties in the copolymer and their interaction with [EMI][TFSI], totally different viscoelastic behaviors were observed. For example, the $G'$ value of 92-AAHA-IL remained higher than $G''$ and reached a plateau regime, supporting that 92-AAHA-IL corresponded to a viscoelastic solid. This feature is presumably due to the ionic association related to the charged pendant groups, similarly serving as physical crosslinks. On the other hand, 90-DAHA-IL could not form a stable network structure and behaved like a viscous liquid with a simple decay of the moduli profiles with decreasing frequency.

The interaction between the charged ammonium group and the other [EMI][TFSI] ion pair also affected the segmental relaxation behaviors of 92-AAHA and 90-DAHA in the ionoconductors. The rheological difference was evaluated by calculating the relaxation time ($\lambda$) using the following equation[39]: $J' = G'/([\eta^*]\omega)^2 = \lambda/[\eta^*]$, where $J'$, $G'$, and $[\eta^*]$ represent the storage compliance, storage modulus, and complex viscosity, respectively. The segmental relaxation time ($\tau_s$) was extracted from the $\lambda$ value at 0.05 rad s$^{-1}$. The added [EMI][TFSI] acted as ion supplier for IC formation and plasticizer. When the content of [EMI][TFSI] was below 30 wt%, it mainly induced a delay in the segmental relaxation through the IC formation. For example, the $\tau_s$ of 92-AAHA-IL with 30 wt% [EMI][TFSI] was 18.8 s, which was much slower than that of neat 92-AAHA ($\tau_s \sim$ 10.3 s, see the red dotted-line in Fig. 1c). However, when the amount of [EMI][TFSI] exceeded 30 wt%, the ionic moieties forming IC in 92-AAHA were saturated, and the excess ionic liquid served as plasticizer. As a result, the opposite effect (i.e., facilitating segmental relaxation) was induced. On the other hand, the $\tau_s$ of neat 90-DAHA (~11.3 s) was continuously reduced upon adding [EMI][TFSI], due to the non-ionic nature of 90-DAHA (Supplementary Fig. 6). In other words, [EMI][TFSI] simply promoted polymer chain oscillations in 90-DAHA-IL irrespective of the composition, similar to conventional plasticizers.

In general, low glass transition temperature ($T_g$) of self-healing materials is considered a crucial requirement to achieve high chain mobility and fast repair[40]. Therefore, 90-DAHA-IL was

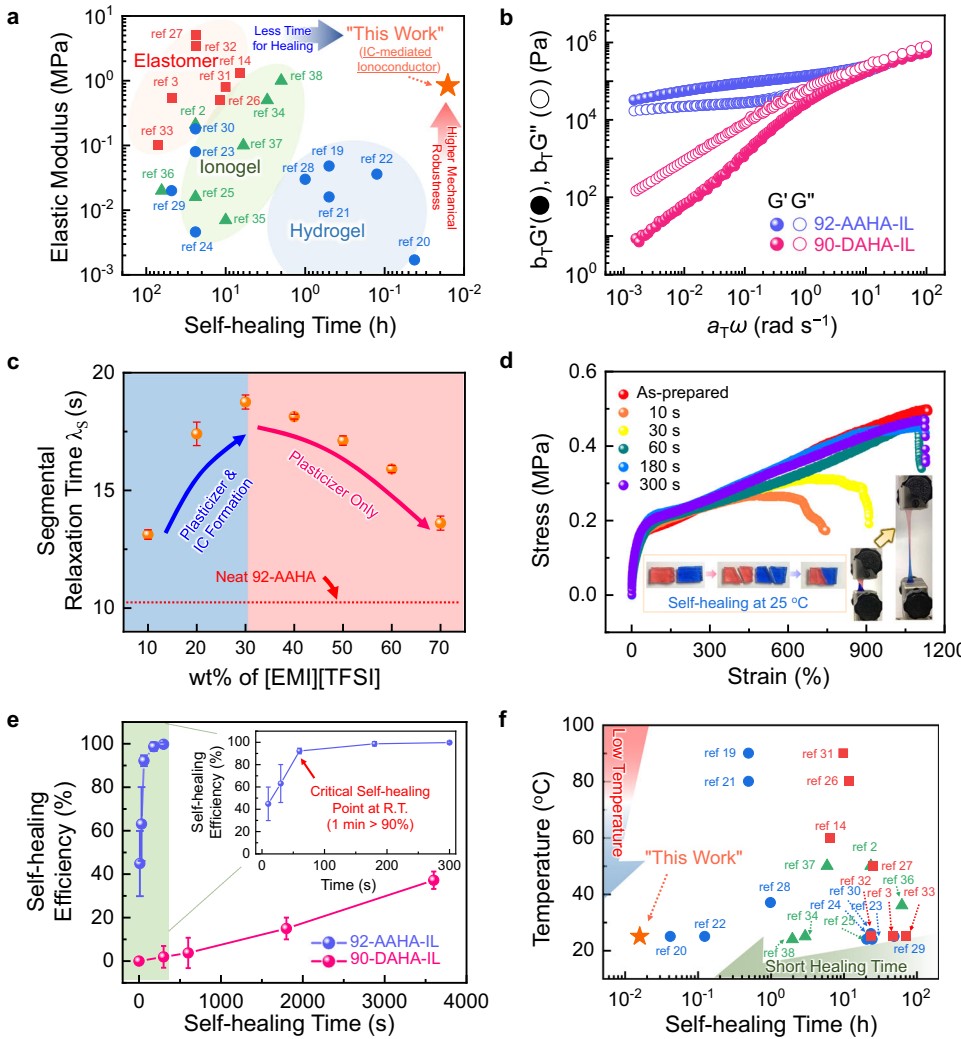

**Fig. 1 Characteristics of highly stretchable and unprecedentedly fast self-healable ionoconductors. a** Trade-off correlation between mechanical robustness and self-healing speed in conventional self-healing systems. **b** Time–temperature master curves of storage ($G'$) and loss ($G''$) moduli for 92-AAHA-IL and 90-DAHA-IL. **c** Influence of weight fraction of introduced [EMI][TFSI] on segmental relaxation time of 92-AAHA-IL. The red-dotted line indicates the segmental relaxation time of neat 92-AAHA. Error bars represent standard deviation. **d** Tensile stress-strain curves of 92-AAHA-IL ionoconductor for various self-healing durations at 25 °C after complete cleavage. The inset shows photographs of the instantaneous self-healing process of 92-AAHA-IL. **e** Plots of self-healing efficiency (%) of 92-AAHA-IL and 90-DAHA-IL versus repair time. The efficiency was estimated from the toughness ratio between the healed and original ionoconductor. Error bars indicate standard deviation. **f** Comparison of performances measured in this work and previously reported self-healing systems, expressed in terms of self-healing time and corresponding healing temperature. Squares, circles, and triangles correspond to elastomer-, hydrogel-, and ionogel-based self-healing platforms, respectively.

expected to show a higher self-healing performance, considering its lower $T_g$ (−23.5 °C) compared to 92-AAHA-IL (−17.6 °C) (Supplementary Fig. 7). Nonetheless, ultrafast (<1 min) self-healing performance was achieved even at 25 °C with the higher-$T_g$ 92-AAHA-IL. The detailed self-healing behaviors were examined by recording stress-strain curves over different healing periods (Fig. 1d). The as-prepared 92-AAHA-IL before being cut into two pieces was highly elastic with superior stretchability upto ~1130% and excellent mechanical stability (Supplementary Fig. 8). The longer contact duration (i.e., healing time) between the two pieces of 92-AAHA-IL allowed ionic species to form more ICs through ion rearrangement, and thus the original properties of the ionoconductor could be restored to a higher extent. Interestingly, the time scale of the self-healing process was dramatically shorter even at low temperatures (e.g., 25 °C) compared to previously reported systems. For example, ~93.1% and ~96.6% of the original yield strength and stretchability, respectively, were recovered in 60 s for 92-AAHA-IL. The immediate self-healing

ability of 92-AAHA-IL was even sufficient to prevent leakage of tiny gas molecules, such as nitrogen (~364 pm), after repair (Supplementary Movie 1 and Supplementary Fig. 9). On the contrary, when the copolymer contained no ionic moieties (namely, 90-DAHA-IL), the stretchability and self-healing ability of resulting ionoconductors were significantly degraded (Supplementary Fig. 10).

The higher performance of 92-AAHA-IL could be illustrated more clearly by comparing the time-depedent self-healing efficiencies, calculated as the toughness ratio between the repaired and original ionoconductors (Fig. 1e). For example, healing for 1 min at 25 °C was sufficient for 92-AAHA-IL to recover its original mechanical properties with high efficiency (~90.3%). When the repair proceeded for 3 and 5 min, the healed ionoconductor exhibited efficiencies of ~98.7 and ~99.7%, respectively. On the other hand, 90-DAHA-IL did not recover its original state even after healing for 1 h, and the achieved efficiency was only ~37.2%. Figure 1f compares the performances of 92-AAHA-IL and

reported self-healing systems based on various mechanisms, such as photoisomerization, hydrogen bonds, ion-dipole interactions, imine bonds, electrostatic interactions, disulfide bonds, and their combination (see also Supplementary Table 2)[2,3,14,19–38]. To the best of our knowledge, the present self-healing ionoconductor ranks at the top in terms of the most important metrics to evaluate the self-healing performance: healing time (~60 s) and process temperature (25 °C).

**Unveiling the ion-cluster-mediated ultrafast self-healing mechanism.** Understanding the importance of the [EMI][TFSI] additive and charged moieties in copolymers is essential to reveal the origin of the ultrafast and highly efficient self-healing performance of 92-AAHA-IL. To this end, we first replaced [EMI][TFSI] in 92-AAHA-IL with a non-ionic additive, propylene carbonate (PC). The weight fraction of PC in the conductor (92-AAHA-PC) was selected as 20 wt% to obtain a similar $T_g$ to 92-AAHA-IL (see Supplementary Fig. 7), so that the effect of $T_g$ on the self-healing behavior could be eliminated. As a result, 92-AAHA-PC exhibited a much lower elastic modulus ($E$ ~0.12 MPa) and self-healing efficiency (~70.3% after 1 h) (Supplementary Fig. 11) compared to 92-AAHA-IL ($E$ ~0.84 MPa and ~90.3% efficiency after 1 min). These results confirm that electrostatic interactions between the charged moieties of copolymers and ionic liquids are needed to achieve high mechanical robustness and self-healing ability. The fraction of P[AA][TFSI] in the AAHA copolymer is another important factor that affects the ion cluster-mediated contribution and overall healing performance. Thus, we prepared two additional ionoconductors based on (co)polymers of 100-AAHA and 61-AAHA including different amounts of P[AA][TFSI]. The self-healing efficiency of 100-AAHA-IL with P[AA][TFSI] homopolymers was poorer than that of 92-AAHA-IL, despite its largest number of sites (i.e., ammonium groups) for interacting with [EMI][TFSI] (Supplementary Fig. 12a) and lowest $T_g$ (Supplementary Fig. 12b). On the other hand, when more PEHA units capable of forming hydrogen bonds were incorporated into AAHA, the contribution of hydrogen bonding to self-healing increased. As a result, the ionoconductor (e.g., 61-AAHA-IL) became stiffer with increasing $T_g$ (Supplementary Fig. 12b, c). The stiffer 61-AAHA-IL was not suitable for rapid self-healing systems, owing to the lower chain mobility (Supplementary Fig. 12a). Therefore, we can conclude that the balanced synergistic effect of the ion-cluster-mediated process and hydrogen bonding is critical to achieve a superior overall self-healing performance, leading to the 92-AAHA-IL being the optimal choice.

Next, we directly verified the presence of interactions between ionic species of charged ammonium groups of the copolymers and [EMI][TFSI] through spectroscopic measurements. Figure 2a shows the difference in the $^1$H nuclear magnetic resonance (NMR) spectrum of 92-AAHA-IL compared to those of neat 92-AAHA and [EMI][TFSI]. The protons ($H_a$) on the trimethyl ammonium group of neat 92-AAHA were detected at $\delta$ ~ 3.33 ppm. After adding 30 wt% [EMI][TFSI], the peak for $H_a$ shifted to ~3.31 ppm due to the interaction with negatively charged [TFSI]$^-$ of [EMI][TFSI]. We note that [EMI][TFSI] alone displays no signal within the chemical shift range of 3.20–3.50 ppm. In contrast, in the range of 7.50–9.50 ppm, where no peaks related to 92-AAHA appeared, an obvious change in peak position for [EMI]$^+$ was observed upon varying the 92-AAHA content in the ionoconductors (Supplementary Fig. 13). The results suggest that both [EMI]$^+$ and [TFSI]$^-$ interacted with 92-AAHA. $^{19}$F NMR (Fig. 2b) and Fourier-transform infrared (FTIR) spectra (Fig. 2c) exhibited similar trends. As the content of [EMI][TFSI] increased, the characteristic peaks arising from

[TFSI]$^-$ notably shifted. On the other hand, no significant peak shift was observed in the spectra of non-ionic 90-DAHA-IL, irrespective of the [EMI][TFSI] amount (Supplementary Fig. 14).

To investigate the molecular dynamics of ionoconductors, dielectric relaxation spectroscopy (DRS) measurements were performed at various temperatures and frequencies. Figure 2d compares the static dielectric constant ($\varepsilon_s$) at 283 K, defined as the low-frequency plateau of the dielectric permittivity $\varepsilon'(\omega)$ before the onset of electrode polarization (EP) using the following Eq. (1):

$$\varepsilon_s \approx \lim_{\omega \to 0}\left[\varepsilon'_{\alpha_2}(\omega)\right] + \varepsilon_\infty \tag{1}$$

The $\varepsilon_s$ values of the ionoconductors were determined as the sum of the frequency-dependent contributions from the ion exchange process (referred to as $\alpha_2$ relaxation) $\varepsilon'_{\alpha_2}(\omega)$ between the ICs present in the ionoconductors and the high-frequency dielectric constant $\varepsilon_\infty$ (see solid lines in Fig. 2d, Supplementary Fig. 15)[41,42]. The 92-AAHA-IL had a lower $\varepsilon_s$ value (~30) than 90-DAHA-IL (~40). Such ordering of the $\varepsilon_s$ (i.e., $\varepsilon_{s,\text{92-AAHA-IL}} < \varepsilon_{s,\text{90-DAHA-IL}}$) remained the same over the entire experimental temperature range (see inset of Fig. 2d), which is attributed to the effective association between the charged ammonium groups in 92-AAHA with [EMI]$^+$ and [TFSI]$^-$ to form ICs. The lower $\varepsilon_s$ of 92-AAHA-IL can be explained by considering that $\varepsilon_s$ is proportional to the product of the square of the dipole moment $\mu$ and the number density of dipoles $\nu$ ($\varepsilon_s \sim \mu^2 \nu$), and that the formation of ICs may be similar to that of near-zero $\mu$ quadrupoles[43].

The reorientation of molecular dipoles associated with the ICs was further investigated through the characterization of dielectric derivative spectra (Eq. (2))[42]:

$$\varepsilon_{\text{der}}(\omega) = (-\pi/2)\left[\partial\varepsilon'(\omega)/\partial \ln \omega\right] \tag{2}$$

in which the conductivity contribution to the loss spectra $\varepsilon''(\omega)$ was eliminated, and then the dipolar relaxation processes were analyzed (Supplementary Fig. 16). Figure 2e exhibits representative $\varepsilon_{\text{der}}(\omega)$ spectra at 283 K for 92-AAHA-IL and 90-DAHA-IL. The $\varepsilon_{\text{der}}(\omega)$ profile was fitted using a combination of a power law for EP and the Havriliak–Negami (HN) function (Eq. (3))[44]:

$$\varepsilon_{\text{der}} = A\omega^{-s} - \frac{\pi}{2}\left[\frac{\partial\varepsilon'_{\text{HN}}(\omega)}{\partial \ln \omega}\right]_{a_2} \text{ with } \varepsilon'_{\text{HN}}(\omega) = \text{Real}\left\{\frac{\Delta\varepsilon}{\left[1 + (i\omega/\omega_{\text{HN}})^a\right]^b}\right\} \tag{3}$$

wherein $A$ and $s$ are constants, $\Delta\varepsilon$ is the relaxation strength, $a$ and $b$ represent shape parameters, and $\omega_{\text{HN}}$ is the characteristic frequency related to the frequency maximal loss $\omega_{\text{max}}$. From the HN fit, we could extract the relaxation frequency maxima $\omega_{\text{max}}$ using Eq. (4)[45]:

$$\omega_{\text{max}} = \omega_{\text{HN}}\left(\sin\frac{a\pi}{2+2b}\right)^{1/a}\left(\sin\frac{ab\pi}{2+2b}\right)^{-1/a} \tag{4}$$

In Fig. 2e, both 92-AAHA-IL and 90-DAHA-IL clearly showed a relaxation (see dashed curves), referred to as the $\alpha_2$ process[46]. Interestingly, despite its higher $T_g$, the $\alpha_2$ process of 92-AAHA-IL was faster than that of 90-DAHA-IL (i.e., $\omega_{\alpha_2}^{\text{92-AAHA-IL}} > \omega_{\alpha_2}^{\text{90-DAHA-IL}}$) (see vertical dashed lines and arrow in Fig. 2e). A different $\alpha_2$ relaxation frequency was observed in the temperature ranges near each $T_g$ (inset of Fig. 2e). Namely, the IC reorientation induced by ion rearrangement ($\alpha_2$ process) was still faster in 92-AAHA-IL even excluding the $T_g$ effect. The temperature dependence of $\omega_{\alpha_2}$ was well-fitted by the Vogel–Fulcher–Tammann (VFT) Eq. (5), shown as solid curves in the inset of Fig. 2e[47]:

$$\omega_{\text{max}} = \omega_\infty\exp\left(-\frac{DT_0}{(T - T_0)}\right) \tag{5}$$

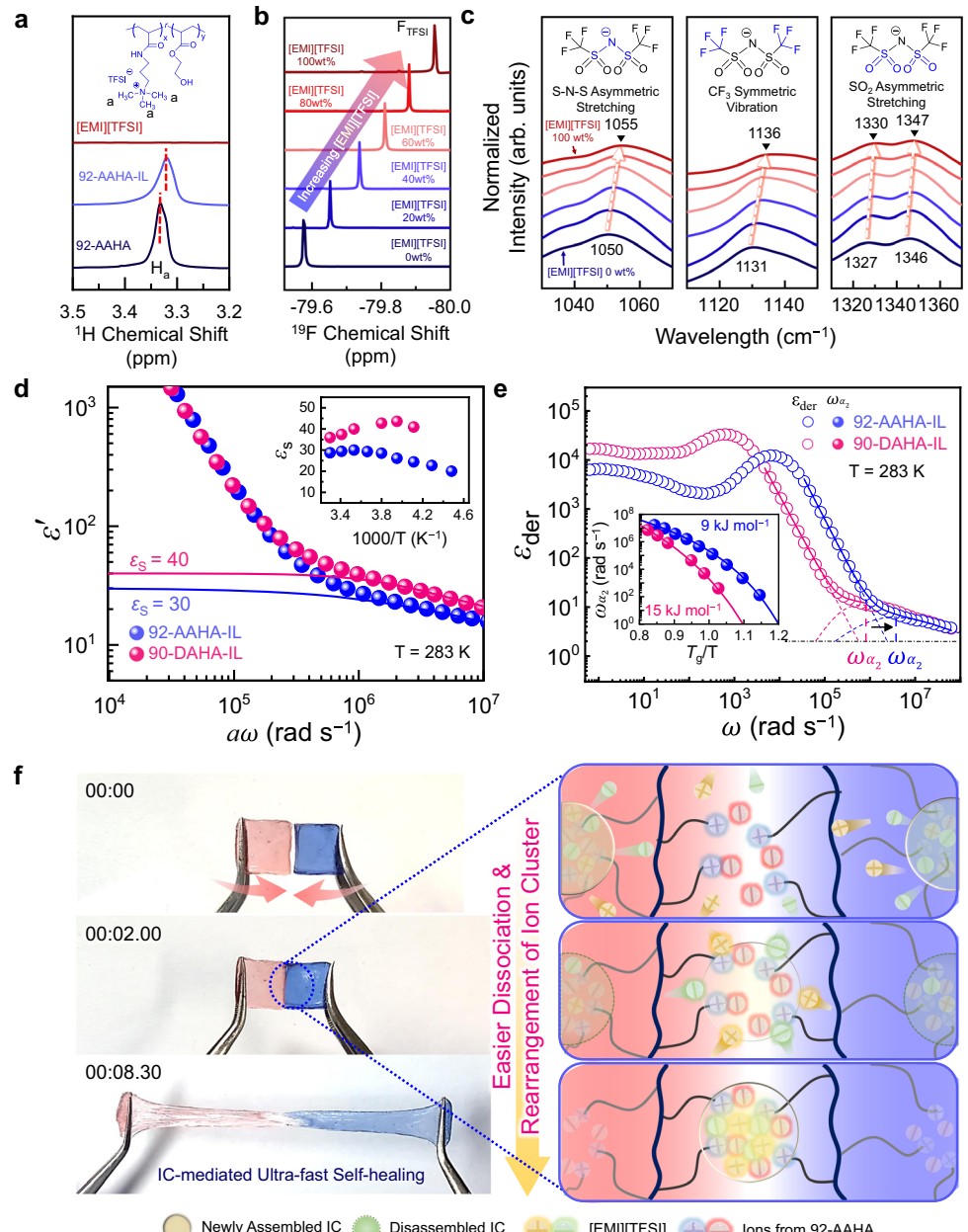

**Fig. 2 Spectroscopic analysis of ion cluster formation for ultra-fast self-healing. a** Comparison of $^1$H NMR spectra for 92-AAHA, 92-AAHA-IL, and [EMI][TFSI], in which 92-AAHA-IL contained 30 wt% [EMI][TFSI]. Changes in **b** $^{19}$F NMR spectra and **c** FTIR absorption peaks related to [TFSI]$^-$ contained in 92-AAHA-IL. The contents of [EMI][TFSI] were 0 (navy line, neat 92-AAHA), 20, 40, 60, 80, and 100 wt% (red line, neat [EMI][TFSI]). **d** Dielectric permittivity spectra ($\varepsilon'$) shifted by horizontal shift factor ($a$) ($a = 1$ for 92-AAHA-IL, $a = 3.4$ for 90-DAHA-IL). Blue and red solid lines represent the static dielectric constant ($\varepsilon_s$) of 92-AAHA-IL and 90-DAHA-IL, respectively. The inset shows $\varepsilon_s$ variations as a function of temperature. **e** Frequency dependence of derivative $\varepsilon_{der}$ spectra of 92-AAHA-IL and 90-DAHA-IL at 283 K. The inset shows the temperature-dependent $\alpha_2$ relaxation frequency with the Vogel-Fulcher-Tammann (VFT) fitting. **f** Schematic illustration of the self-healing mechanism of 92-AAHA-IL, based on the rapid dissociation and re-assembly of ion clusters (ICs).

where $\omega_\infty$ is the high-temperature limiting frequency, $T_0$ is the Vogel temperature, and $D$ is the strength parameter (listed in Supplementary Table 3). The VFT fit parameters allowed us to estimate the activation energy of the IC reorientation ($DT_0 \approx E_a/R$). The estimated energy barrier for ion rearrangement of 92-AAHA-IL ($E_a = 9 \text{ kJ mol}^{-1}$) was 40% lower than that of 90-DAHA-IL ($E_a = 15 \text{ kJ mol}^{-1}$). The lower energy barrier is presumably due to the dissociation-association of the IC groups through the intermolecular interactions detected in the NMR (Fig. 2a, b) and FTIR (Fig. 2c) spectra. The faster $\alpha_2$ relaxation of

92-AAHA-IL was also reflected in its higher ionic conductivity compared to that of the lower $T_g$ 90-DAHA-IL (Supplementary Fig. 17). These results indicate that easy ion dissociation/association and thus rapid IC reorientation are the key factors for the ultrafast self-healing behaviors (Fig. 2f and Supplementary Movie 2). The importance of a facile ion dissociation/association was further supported by the results of experiments performed with various ionic liquid additives (Supplementary Fig. 18), in which we varied the cations and anions of the ionic liquid. However, the fastest recovery was still observed with [EMI][TFSI], which can be

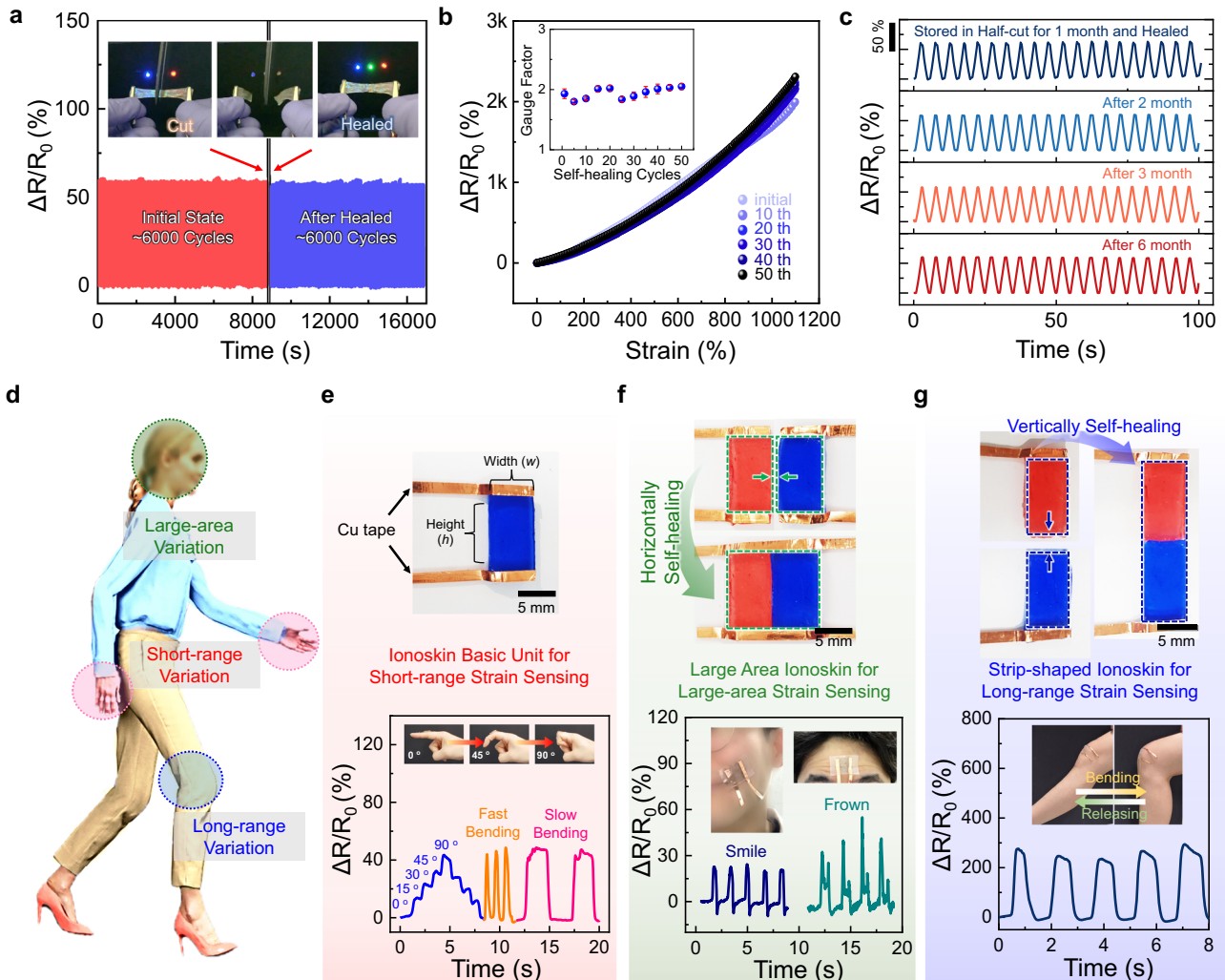

**Fig. 3 Mechanical/electrical self-healing of ionoconductors and applications in area-adjustable ionoskins. a** Cyclic stability tests of 92-AAHA-IL-based ionoconductor during 6000 initial cycles before mechanical damage and 6,000 additional cycles after being healed. The inset images represent the state of the ionoconductor during cutting and healing through the on/off state of LEDs. **b** Strain-dependent resistance changes ($\Delta R/R_0$) during cut and healing cycles. The inset indicates the variation in gauge factor as a function of self-healing iterations. Error bars indicate standard deviation. **c** Stable IC-mediated self-healing process of 92-AAHA-IL, for which two pieces of the ionoconductor were kept under ambient conditions for up to 6 months and healed prior to the cyclic test. **d** Schematic illustration of various cases according to strain variation and range when applying to ionoskins. **e** Image of ionoskin basic unit colored blue for easy identification (top) and tracking of various finger motions (bottom). **f** Large-area ionoskin prepared by self-healing along the height direction (top) to monitor facial skin deformations (bottom). **g** Fabrication of strip-shaped ionoskin (top) and its use for detecting long-range strain variations (bottom).

explained by its highest molar conductivity related to ion–ion dissociation and ion movement.

**Recovery of electrical properties of ionoconductors via self-healing and application in area-adjustable ionoskin sensors.** To apply self-healable ionoconductors in electrochemical systems, the range of healable features should include not only mechanical but also electrical properties[48]. In particular, to explore their use in ionoskin strain sensors, we evaluated the recovery of electrical characteristics by recording the change in resistance during cyclic uniaxial tensile tests (Fig. 3a). As 92-AAHA-IL was stretched, the overall resistance increased, corresponding to a relative resistance ($\Delta R/R_0$) of ~60.2%. During the first 6000 stretching/releasing cycles, a similar $\Delta R/R_0$ was maintained without notable degradation. Then, we cut the ionoconductor in half to break the electric circuit. The open-circuit state was visualized through the on/off state of an LED bulb connected to the ionoconductors and power source in series (see inset of Fig. 3a). When the two

pieces of the ionoconductor came into contact and self-healed, the electrical path was restored and the current level became constant within ~1.8 s (Supplementary Fig. 19). The long-term stability of the healed 92-AAHA-IL remained high over 6000 additional cycles.

The sensory characteristics of 92-AAHA-IL were evaluated in detail. The 92-AAHA-IL successfully distinguished applied mechanical strain with high precision. For example, a change in $\Delta R/R_0$ (~5.7%) was clearly shown even with a relatively small applied strain of 10% (Supplementary Fig. 20a). Upon severe deformation of the ionoconductor, the overall resistance further increased. In particular, the measured $\Delta R/R_0$ was linearly proportional to the applied strain (Supplementary Fig. 20b), which is one of the important features of strain sensors. The gauge factor (GF) was extracted from the slope of a linear fit to the $\Delta R/R_0$ versus applied strain plot, giving a GF of ~1.92. To examine the effect of the self-healing process on the sensory performance, we recorded the strain-dependent $\Delta R/R_0$ as a

function of the number of self-healing iterations (Fig. 3b). No degradation of the electrical properties of 92-AAHA-IL was observed even after 50 self-healing cycles. As a result, the GF value was not affected by the repeated repair of mechanical defects of the ionoconductor (see inset of Fig. 3b). More interestingly, the IC-mediated self-healing mechanism was still valid even 6 months after cutting the ionoconductor in half (Fig. 3c), which was attributed to the non-volatility of 92-AAHA-IL even without encapsulation. This excellent longevity highlights a high potential for applications where very durable and stable ionoconductors are necessary.

The most straightforward sensing applications include skin-type wearable strain sensors, referred to as ionoskins, which can monitor a variety of human movements[49]. One requirement for accurate motion detection is using appropriately sized ionoconductors to cover the area where mechanical deformation is induced (Fig. 3d). For example, large-area ionic conductors need to be applied when strain changes occur over a wide area (i.e., a facial expression). Long strip-shaped ionic conductors are suitable to synchronize longitudinal variations (i.e., bending motions of arms and legs). Taking advantage of its ultrafast self-healing properties, the dimensions of 92-AAHA-IL can be readily tailored, depending on the purpose by cutting or re-assembling the film. As a basic unit, we used an ionoconductor film with dimensions of 10 mm (height, $h$) and 6.5 mm (width, $w$) (top image of Fig. 3e) and attached it to a finger joint. Such a small film effectively tracked the angular motion and bending speed of the finger through changes in $\Delta R/R_0$ (bottom image of Fig. 3e). To monitor motions occurring over a large-area motion, two basic ionoconductor units can be self-healed along the height direction to conveniently make a large-area ionoskin ($h$: 10 mm, $w$: 13 mm) (top image of Fig. 3f). Deformations generated on the skin upon grimacing or smiling were successfully recognized with high precision (bottom image of Fig. 3f). On the other hand, self-healing of two basic ionoconductor fragments along the width direction produced a strip-shaped ionoskin ($h$: 20 mm, $w$: 6.5 mm) capable of sensing long-range and large strain changes (Fig. 3g). The demonstration represents the benefits of the extremely fast self-healing characteristics of 92-AAHA-IL for area-adjustable ionoskins.

**Deformable electronics platforms: User-reconfigurable AC electroluminescent displays**. We propose a deformable electronic platform of reconfigurable ACEDs that can be readily dis-assembled and re-assembled in accordance with the user's demand by exploiting the superior self-healing ability of 92-AAHA-IL. ACEDs were selected as optimal electronic devices for maximizing the advantages of ultrafast self-healing, because the ionoconductors themselves can serve as electrodes, different from typical electronic components (Supplementary Fig. 21a)[50]. In addition, the high stretchability (Fig. 1d) and optical transparency (Supplementary Fig. 22) of 92-AAHA-IL enhance its applicability to deformable displays. ACEDs were straightforwardly fabricated by sandwiching an emissive layer containing either ZnS:Cu (green) or ZnS:Cu,Al (blue) between two self-healable 92-AAHA-IL films (Supplementary Fig. 21b). The detailed device characteristics (e.g., voltage-dependent luminance and CIE color coordinates, electroluminescent (EL) spectra, and frequency-dependent luminance) are given in Supplementary Fig. 23.

When a symmetric square wave (peak-to-peak voltage of 200 $V_{pp}$ at 25 kHz) was applied to the device containing ZnS:Cu, a green-colored light emission was observed (Fig. 4a). Then, the device was cut into two components using a razor blade. When the external bias was applied through two electrodes installed on the edge of the device, each unit successfully emitted light. To exploit the immediate self-healing of the electrodes (92-AAHA-IL

ionoconductors), the two individual devices were re-contacted along the cross-section to be healed. Although the emissive layer did not have self-healing ability, the resulting ACED was repaired through the self-healing of the two electrodes and operated similar to the original one. For example, no significant difference in luminance was detected even after several consecutive cutting/healing processes (Fig. 4b).

The reconfigurable ACEDs enabled the fabrication of a dual-color emitting device by assembling green and blue emitting units via self-healing (Fig. 4c and Supplementary Movie 3). When the cells were combined into one device and an AC input was supplied through two electrodes located on the opposite cells, both the blue and green cells operated independently without noticeable interference. More interestingly, the healed device exhibited high deformability, leading to successful operation under various deformations, including bending, twisting, and stretching (Fig. 4c). We further expanded our reconfigurable electronics strategy, for which nine different ACEDs were assembled into a $3 \times 3$ pixelated display (Fig. 4d). We note that pixelation using self-healing has not been reported before; this approach dramatically simplified the overall fabrication process compared to conventional systems[51]. When connected to an AC power source through two pieces of Cu tapes placed on the corners, all cells were turned on simultaneously. The $3 \times 3$ pixelated multicolor ACED exhibited excellent deformability even when crumpled and stretched (Fig. 4d and Supplementary Movie 4), indicating the highly promising potential of reconfigurable and deformable electronics for a variety of practical uses.

## Discussion

In summary, the trade-off between mechanical properties and self-healing performance in conventional healable materials was overcome by developing the mechanically robust, ultra-fast self-healing 92-AAHA-IL, and a concept of reconfigurable electronics was successfully demonstrated. The IC-driven self-healing mechanism and its effects on the physical properties of ionic conductors were investigated in-depth by rheometry and spectroscopy. We concluded that (1) the produced ICs served as physical crosslinking points, (2) an optimal content (30 wt%) of [EMI][TFSI] should be introduced to 92-AAHA for maximizing the self-healing performance (<1 min at 25 °C) with high mechanical robustness (elastic modulus ~0.84 MPa), and (3) fast rearrangement of ICs is the origin of the ultra-fast self-healing ability. The ultimate self-healing characteristics of 92-AAHA-IL were successfully exploited to develop deformable electronics (e.g., area-adjustable ionoskin sensors and user-reconfigurable ACEDs). Very interestingly, the 92-AAHA-IL could be successfully repaired in extreme environments such as sub-zero temperatures (Supplementary Fig. 24a and Movie 5) and underwater conditions (Supplementary Fig. 24b and Movie 6). Moreover, the high thermal stability (Supplementary Fig. 25) and successful application of 92-AAHA-IL in self-healable energy storage devices (e.g., supercapacitors, Supplementary Fig. 26) expanded its practical application range. Overall, the present results highlight the high potential impact of 92-AAHA-IL and provide insights into future electronics platforms such as reconfigurable electronics.

## Methods
**Materials**. (3-Acryloamidopropyl)trimethylammonium chloride solution ([AAHA][Cl]) (75 wt% in H$_2$O), 2-hydroxyethyl acrylate (2-HEA, 96%), 1-ethyl-3-methylimidazolium bromide ([EMI][Br]; >97.0%), 2,2'-azobis(2-methylpropiona-midine) dihydrochloride (V50, 97%), N,N-dimethylacetamide (DMAc, 99.8%), 1-butyl-1-methylpyrrolidinium bis(trifluoromethylsulfonyl)imide ([P$_{14}$][TFSI], >98.0%), butyltrimethylammonium bis(trifluoromethylsulfonyl)imide ([N$_{4111}$][TFSI], 99%), 1-butyl-3-methylimidazolium bis(trifluoromethylsulfonyl)imide ([BMI][TFSI], ≥98.0%), 1-butyl-3-methylimidazolium hexafluorophosphate ([BMI][PF$_6$], ≥97.0%), 1-butyl-3-methylimidazolium tetrafluoroborate ([BMI]

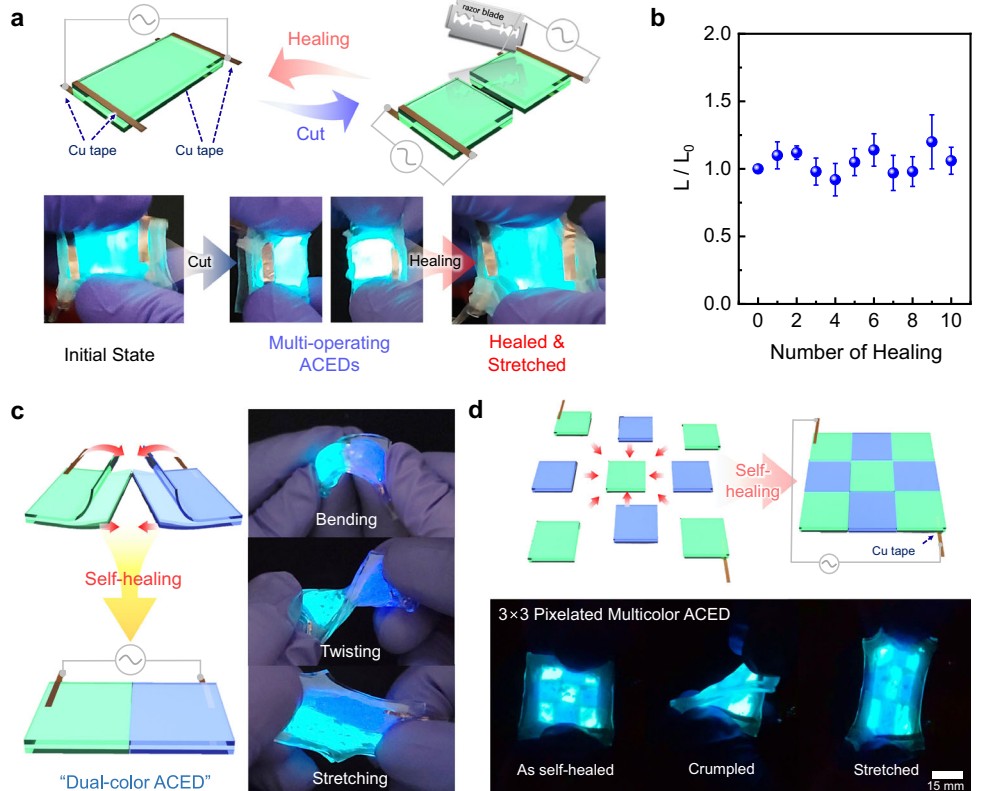

**Fig. 4 Reconfigurable AC electroluminescent displays (ACEDs). a** Schematic illustrations and photographs of the cutting/healing process of an ACED based on ultrafast self-healing ionoconductors. **b** Changes in luminance during consecutive cutting/healing cycles. Error bars indicate standard deviation. **c** Illustrations and photographs of dual-color-emitting ACEDs fabricated by assembling green- and blue-emitting ACEDs via self-healing. **d** Strategy to fabricate the 3 × 3 pixelated multicolor ACED and photographs illustrating its successful operation under various deformations, such as crumpling and stretching.

[BF$_4$], ≥98%), indium tin oxide (ITO) (<100 nm sized particles, 30 wt% in iso-propanol), and 2,2′-azobis(2-methylpropionitrile) (AIBN, 98%) were purchased from Sigma-Aldrich. N-[3-(Dimethylamino)propyl]acrylamide (DA, >98.0%) was obtained from Tokyo Chemical Industry Co., Ltd. Lithium bis(tri-fluoromethanesulfonyl)imide ([Li][TFSI]) and 3 M VHB elastomer were obtained from 3 M (Minnesota, USA). Ether (99.0%), methyl alcohol (99.5%), and 2-propanone (99.5%) were purchased from Samchun Pure Chemical Co., Ltd. (Pyeongtaek-si, South Korea). Deionized (DI) water (HPLC) was purchased from J.T.Baker (New Jersey, USA). Ecoflex 00–30 was purchased from Smooth-On (Macungie, Pennsylvania, USA). Green (D512S, ZnS:Cu), and blue (D417S, ZnS:Cu,Al) electroluminescent materials were obtained from Shanghai KPT Co., Ltd. The [EMI][TFSI] ionic liquid was prepared via an anionic exchange reaction between [EMI][Br] and excess [Li][TFSI] in deionized (DI)-water at 60 °C.

**Characterizations**. The number average molecular weight ($M_n$) and polydispersity (Đ) of the obtained copolymers were measured using size exclusion chromatography (SEC, LC-4500, JASCO) with a refractive index detector (RI-4030, JASCO) calibrated with standard poly(ethylene oxide) (calibration kit, Scientific Polymer Products, Inc.). $^1$H and $^{19}$F NMR spectra were recorded on an Avance III HD500 instrument using acetone-$d_6$ (CD$_3$COCD$_3$, 99.9%, Sigma-Aldrich) as a solvent. FTIR spectra were recorded using the attenuated total reflection (ATR) technique (Spectrum 100, PerkinElmer). DRS measurements were carried out with a Novocontrol GmbH Concept 40 broadband dielectric spectrometer. Samples (thickness: 0.05–0.1 mm) were prepared by sandwiching ionoconductors between freshly polished brass electrodes and then placed in a spectrometer equipped with a Quatro Cryosystem sample chamber, a vacuum-isolated cryostat, and a nitrogen line. All measurements were performed at a sinusoidal AC voltage with an amplitude of 0.1 V over the frequency range of $10^{-1}$–$10^7$ Hz. Data were collected with isothermal frequency sweeps every 10 K from 313 to 223 K. The $T_g$ values of the prepared ionoconductors were measured by differential scanning calorimetry (DSC, DSC 4000, PerkinElmer) under N$_2$ atmosphere. The sample (typical weight ~6.0 mg) was first heated to 150 °C and held at this temperature for 20 min to erase its thermal history, followed by quenching at −30 °C. Then, DSC thermograms were recorded during a second heating stage at a rate of 10 °C min$^{-1}$. The dynamic mechanical behaviors were characterized by an oscillatory rheometer (MCR-92, Anton Paar) with 8 mm parallel plates. The sample gap was fixed at ~1 mm and the strain was set to a small strain amplitude of 1% (linear response) for frequency-dependent measurements.

Mechanical properties were characterized using a z-axis motorized force applier (ESM303, Mark-10) and a force gauge (M5-10, Mark-10). Tensile strain−stress curves were recorded at a rate of 10 mm min$^{-1}$ for uniform rectangular samples (thickness: 300 μm, width: 10 mm, length: 30 mm). A UV–vis spectrometer (V-730, JASCO) was used to demonstrate the high transparency of the ionoconductors, with a scan range and rate of 400–1100 nm and 400 nm min$^{-1}$, respectively. The changes in capacitance of the ionoconductor and emissive layer were monitored with an LCR meter (E4980AL, Keysight Technologies) at 1 V. Conductivities were measured using electrochemical impedance spectroscopy (EIS; IM6, Zahner) at an amplitude of 10 mV and in the frequency region from $10^6$ to $10^{-1}$ Hz.

**Synthesis of 92-AAHA copolymer**. A mixture containing the [AAHA][Cl] (52.82 g, 255.54 mmol) and 2-HEA (0.3 g, 2.58 mmol) monomers and the V-50 initiator (1.00 mg, 0.0037 mmol) was placed in a two-neck flask. After purging with Ar gas at room temperature for 1 h, free radical polymerization was conducted at 80 °C for 2 h in an oil bath, followed by quenching with liquid nitrogen for ter-mination. The resulting solution was precipitated by pouring it into excess 2-propanone. The collected copolymer was dried at 60 °C in vacuum. The deter-mined $M_n$ and Đ values of the obtained copolymer were 490,000, and 2.6, respectively. To replace [Cl]$^-$ with [TFSI]$^-$, excess [Li][TFSI]-containing water solution was poured into an aqueous polymer solution at ambient conditions. The precipitated 92-AAHA was quickly collected and dried at 60 °C in vacuum. The resulting copolymer was re-dissolved in acetone and poured into DI water to remove unreacted excess [Li][TFSI]. These steps were repeated five times for purification. The other polymers were synthesized using a similar procedure. The P[AA][TFSI] mole fraction in the copolymers was adjusted by controlling the feed ratio of monomers.

**Synthesis of 90-DAHA copolymer**. A mixture solution including DA (65.93 g, 422.02 mmol), 2-HEA (0.50 g, 4.27 mmol), and AIBN initiator (1.00 mg, 0.0061 mmol) was introduced into a two-neck flask with 5 mL of DMAc. Free radical polymerization of the monomer mixture was conducted at 80 °C after Ar purging for 1 h. The polymerization was terminated after 10 h by quenching with liquid nitrogen. The solution was poured into an excess amount of ether, leading to precipitation of the crude product. The precipitation process was repeated three times for further purification. After complete drying in vacuum, the molecular

characteristics of 90-DAHA were measured using SEC, giving $M_n$ and $Đ$ of 473,000 g mol$^{-1}$ and 1.9, respectively.

**Preparation of ionoconductors.** Ionoconductors were prepared by physical blending methods as follows. The synthesized copolymers were mixed with a predetermined amount of [EMI][TFSI] using acetone as co-solvent and stirred for 3 h at 80 °C. The homogeneously mixed solution was poured onto a teflon mold. After fully drying the co-solvent in vacuum, transparent ionoconductor films were obtained.

**Fabrication and characterization of area-adjustable strain sensors.** To fabricate self-healable ionoconductor-based area-adjustable strain sensors, the 92-AAHA-IL solution was cast onto rectangular-shaped templates (thickness: 300 μm, width: 10 mm, length: 30 mm), followed by drying at 100 °C in vacuum. Changes in resistance of the ionoconductor under various strains were monitored using a source meter (2420, Keithley, USA). To verify the feasibility of a skin-type strain sensor (referred to as ionoskin), the ionoconductors were attached to several body parts of a volunteer who participated in the experiment with fully informed consent.

**Fabrication and characterization of reconfigurable ACEDs.** To demonstrate the new class of reconfigurable ACEDs, the square-shaped 92-AAHA-IL ionoconductor (thickness: 300 μm, width: 1 mm, and length: 1 mm) was fabricated as an electrode via the method described above. A green emissive layer was made by introducing 0.3 g cm$^{-3}$ ZnS:Cu microparticles into a mixture of Ecoflex 00–30 A and Ecoflex 00–30 B. The prepared ZnS:Cu/Ecoflex composite was spin-cast on a poly(methyl methacrylate) (PMMA)-coated glass at 2000 rpm for 1 min. The obtained film was thermally cured in an oven at 80 °C for 3 h. Then, a 90 μm-thick EL layer was cut into desired shapes and inserted between two ionoconductors. Each ionoconductor was coupled with a piece of copper tape to connect with the external power source. The entire device was then wrapped with VHB tapes and pressed at 80 °C upon application of 40 N force for encapsulation. The emissive spectra and luminance of the ACED were acquired using a spectroradiometer (CS-2000, Konica Minolta). A high-voltage power amplifier (HA-805, Pintek) was used with a function generator (33210A, Keysight) to operate the ACED.

**Fabrication and characterization of self-healable supercapacitors.** To assess the applicability in energy storage systems such as supercapacitors, the 92-AAHA-IL film (thickness: 2 mm, width: 10 mm, and length: 10 mm) was employed as a self-healable electrolyte for an electrical double layer capacitor (EDLC). To enlarge the electrode surface, ITO particles were additionally spin-coated (1200 rpm, 60 s) onto the ITO-coated glass and vacuum-dried at 60 °C. The fabrication of the supercapacitor was completed by sandwiching 92-AAHA-IL between two ITO particle-decorated electrodes. The performance of the EDLC was characterized using cyclic voltammetry (CV, Wave Driver 10, Pine Instrument) and galvanostatic charge-discharge (GCD, WBCS30000L, WonATech) measurements.

## Data availability
All data supporting the findings of this study are available within this article and Supplementary Information files or from the corresponding author upon reasonable request. Source data are provided with this paper.

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

## Acknowledgements
This work was supported by the National Research Foundation of Korea (NRF) Grant funded by the Ministry of Science and ICT for Original Technology Program (NRF-2020M3D1A2102869) (H.C.M.). This work was supported by the National Research Foundation of Korea (NRF) grant funded by the Korea government (MSIT) (NRF-2022R1A2C4001425) (H.C.M.).

## Author contributions
Y.M.K. and H.C.M. conceived the concept and designed this study. Y.M.K. mainly conducted the experiments and analyzed the data. J.H.K. and S.K. assisted the experiments and data analysis. Y.M.K., S.K., U.H.C., and H.C.M. discussed and interpreted the results. All authors wrote the manuscript. H.C.M. supervised the project.

## Competing interests
The authors declare no competing interests.
