## [Peer Review File · Nature Communications]

Ion-Cluster-Mediated Ultrafast Self-Healable Ionoconductors for Reconfigurable ElectronicsREVIEWER COMMENTS

Reviewer #1 (Remarks to the Author):

In this manuscript, the authors developed an ion-cluster-mediated ionoconductor with desirable mechanical properties and ultra-fast self-healing ability with a low temperature, and was used to demonstrate deformable strain sensors and reconfigurable light-emitting electroluminescent displays. This is a very interesting study and is suitable for publication in Nature Communications after addressing following concerns.

1. 92-AAHA-IL showed the highest self-healing performance compared with other materials in Fig.1a. It is confused why the ion-cluster-mediated self-healing is faster than others (e.g., B-O bond, H bond). In the article (Adv. Mater. 2021, 2008479), a nearly identical ionogel was prepared (Poly[MATAC][TFSI]-[N4111][TFSI]) but the self-healing performance was different. Is there other influencing factor?
2. The properties of the ionogel were not characterized very well. The mechanical recoverability, the stability of ionogels and the conductivity should be characterized.
3. Page 8 "However, the self-healing efficiency of 100-AAHA corresponding to the P[AA][TFSI] homopolymer was not the highest despite the highest number of sites (i.e., ammonium groups) for interacting with [EMI][TFSI] and the lowest Tg." Can the authors give more discussion on this point?
4. Can the ultra-fast self-healing ionogel be used in extreme conditions (e.g., low temperature, underwater)?
5. Please double check the spelling and grammar throughout the entire manuscript.

Reviewer #2 (Remarks to the Author):

the paper "Ion-Cluster-Mediated Ultrafast Self-healable Ionoconductors for Reconfigurable Electronics" describes a material with rapid self-healing abilities (<min) at low temperature (RT).

the paper is written as a step-by-step story where each point is addressed to convince the reader on the potentialities of the material ; especially, the synthesis/characterizations of material, their properties, the major role of EMI:TFSI in the self-healing behavior.

According to me the paper deserves to be published in Nature Comm. because the demonstration of superior self-healing is evidenced all along, compared to other categories of materials. Furthermore the broad applications are also addressed showing all potentialities.

here I have some suggestions/comments that authors must reply :

1. EMI:TFSI appear as a "magic" tandem in so many publications. please, explain in the manuscript why use this couple of ions ; is it so specific to the low temp and rapid self-healing abilities? other ionic liquid tandem would have given similar results ? (i.e. other imidazolium, or pyridiniums or phosphoniums etc... with several families of anions of the long list of IL). Can the author provide at least an example that could demonstrate that IC-mediated self-healing is specific to the EMI:TFSI or (on the contrary) could be enlarged to broader family of IL.
2. Figure 1a : the star point "This Work" is really out of any other published studies and seems to prove the breakthrough in the area of self-healing materials. So authors define the current work as a new point of an existing family in the graph or they claim this is a new category of self-healable materials.
3. Application: all examples of application in the manuscript give totally impressive results and videos are impressive too. Now if I look for my own concern, one application, supercapacitor, could have been explored: typically how is evolving the specific power of supercapacitor after 10, 20, 50 self-

healing steps. This is crucial and could be also easily adressed here. (Evolution of specific power would be mainly resulting of ionic behavior during self-healing steps)

Response to Review and Reference to Changes for ncomms-22-08900

Reviewer's comments

Reviewer #1 (Remarks to the Author)

In this manuscript, the authors developed an ion-cluster-mediated ionoconductor with desirable mechanical properties and ultra-fast self-healing ability with a low temperature, and was used to demonstrate deformable strain sensors and reconfigurable light-emitting electroluminescent displays. This is a very interesting study and is suitable for publication in Nature Communications after addressing following concerns.

We thank the reviewer 1 for his/her positive assessment and recommendation of our manuscript. We hope that the revised manuscript describes his/her concerns satisfactorily.

1. 92-AAHA-IL showed the highest self-healing performance compared with other materials in Fig.1a. It is confused why the ion-cluster-mediated self-healing is faster than others (e.g., B-O bond, H bond). In the article (Adv. Mater. 2021, 2008479), a nearly identical ionogel was prepared (Poly[MATAC][TFSI] -[N4111][TFSI]) but the self-healing performance was different. Is their other influencing factor?

Thank you for this invaluable comment. Most self-healing systems are based on the reversible bond-mediated healing process, in which promoting segmental motion of polymer chains is critical to achieve fast self-healing performance. For this purpose, two strategies have been commonly employed : (1) applying external energy (e.g., heating or UV-irradiation) and (2) employing low- T_g polymers. In contrast, the ion-cluster-mediated mechanism does not require significant movement of large polymer chains for healing. Instead, easily movable small ions (namely, [EMI]⁺ and [TFSI]⁻) form/dissociate ion-clusters with charged moieties on the polymer chain. As a result, low-temperature, rapid self-healing is possible. This approach has never been tried before.

As the reviewer 1 pointed out, although ionogels consisting of poly[MATAC][TFSI] (polymer) and [N₄₁₁₁][TFSI] (ionic liquid) look similar to our system (Adv. Mater. **33**, 2008479 (2021)), their self-healing took a longer time (~3 h in ambient condition). The difference in self-healing performance can be rationalized as follows:

1) Firstly, the used ionic liquid and its physical properties are different (i.e., [EMI][TFSI] (this work) versus [N₄₁₁₁][TFSI] (Adv. Mater. **33**, 2008479 (2021))). To facilitate ion-cluster dissociation/association for rapid self-healing, molar conductivity (Λ) should be considered a key ionic liquid characteristic, which correlates with ion-ion dissociation and ion movement. Namely, ionic liquids with higher Λ are preferred for the ion-cluster-mediated mechanism. Considering Walden's rule ($\Lambda\eta = \text{constant}$, where η represents the viscosity of ionic liquid) and η values of 27 and 77 mPa·s for [EMI][TFSI] and [N₄₁₁₁][TFSI], respectively (J. Phys. Chem. B **110**, 19593-19600 (2006)), a much higher Λ is estimated for [EMI][TFSI]. Accordingly, our system favors with [EMI][TFSI] ion-cluster formation/dissociation during the healing process. The correlation between Λ and self-healing performance is further provided in the response to the comment 1 made by the reviewer 2.

2) Another possible parameter contributing to the unprecedentedly superior self-healing of our system is different polymeric molecular characteristics (i.e., P[AA][TFSI]-*r*-PHEA copolymer

(this work) versus poly([MATAAC][TFSI] a homopolymer (*Adv. Mater.* **33**, 2008479 (2021))). Although P[AA][TFSI]-*r*-PHEA contains only ~8 mol% of PHEA capable of hydrogen bonding, the effect was not trivial. As shown in Supplementary Fig. 12a, self-healing ability of P[AA][TFSI]-*r*-PHEA (92-AAHA-IL) (60 s for 90% healing) is much better than that of P[AA][TFSI] homopolymer-based system (100-AAHA-IL) (1000 s for 90% healing). This result implies the synergistic effect of hydrogen bonding through PHEA unit and its importance for high self-healing performance.

Supplementary Fig. 12 | a Differences in self-healing efficiencies depending on the copolymer hosts. Error bars indicate standard deviation.

Change: To address reviewer 1’s comment, the following content has been updated in the revised manuscript, and also the paper (*Adv. Mater.* **33**, 2008479 (2021)) has been referenced additionally.

“The importance of a facile ion dissociation/association was further supported by the results of experiments performed with various ionic liquid additives (Supplementary Fig. 18), in which we varied the cations and anions of the ionic liquid. However, the fastest recovery was still observed with [EMI][TFSI], which can be explained by its highest molar conductivity related to ion-ion dissociation and ion movement.” (line 20-24 on page 10 in the revised manuscript)

[48] Yu, Z. et al. Underwater communication and optical camouflage ionogels. *Adv. Mater.* **33**, 2008479 (2021). (ref. 48 in the revised manuscript).

2. The properties of the ionogel were not characterized very well. The mechanical recoverability, the stability of ionogels and the conductivity should be characterized.

We appreciate the reviewer 1 for this kind suggestion. We further characterized physical properties of the ionoconductor, including thermal stability (TGA thermogram), temperature-dependent ionic conductivity, and mechanical recoverability/stability in terms of recovery ratio and residual strain during stretching/releasing cycles.

Change: To address reviewer 1’s point, the following contents have been updated in the revised manuscript and Supplementary Information.

“Moreover, the high thermal stability of 92-AAHA-IL (Supplementary Fig. 25) and its successful application in self-healable energy storage devices (e.g., supercapacitors, Supplementary Fig. 26) expanded its practical application range.” (line 21-24 on page 14 in the revised manuscript)

Supplementary Fig. 25 | TGA thermogram of the 92-AAHA-IL, indicating its high thermal stability up to ~300 °C.

“The faster α_2 relaxation of the 92-AAHA-IL was also reflected in its higher ionic conductivity compared to that of the lower T_g 90-DAHA-IL (Supplementary Fig. 17).” (line 16-18 on page 10 in the revised manuscript)

Supplementary Fig. 17 | Temperature dependence of ionic conductivity for 92-AAHA-IL and 90-DAHA-IL.

“The as-prepared 92-AAHA-IL before being cut into two pieces was highly elastic with superior stretchability upto ~1130% and excellent mechanical stability (Supplementary Fig. 8).” (line 17-19 on page 6 in the revised manuscript)

Supplementary Fig. 8 | **a** Stress-strain curves during cyclic stretching/releasing at 800% and **b** corresponding changes in recovery ratio and residual strain. The recovery ratio of ~83.2% and residual strain of ~8.6% after 100 consecutive cycles represent the high mechanical reliability of 92-AAHA-IL. Error bars indicate standard deviation.

3. Page 8 “However, the self-healing efficiency of 100-AAHA corresponding to the P[AA][TFSI] homopolymer was not the highest despite the highest number of sites (i.e., ammonium groups) for interacting with [EMI][TFSI] and the lowest T_g .” Can the authors give more discussion on this point?

This comment is important to understand our system. When we compared the 92-AAHA-IL and 100-AAHA-IL at the same [EMI][TFSI] content (e.g., 30 wt%) for fairness, the 92-AAHA-IL indicated much faster healing performance (see Fig. 1e and Supplementary Fig. 12a). The only difference between two systems is the inclusion of ~8 mol% PHEAs capable of hydrogen bonding in copolymers, indicating the importance of the synergistic effect of ion-cluster-mediated process and hydrogen bonding. We also note that the balance between those two self-healing mechanisms is crucial for maximizing overall self-healing performance. For example, when the content of PHEAs in copolymers increased to ~39 mol% (namely, 61-AAHA-IL), the contribution of hydrogen bonding to self-healing increased and the resulting gel became stiffer, leading to much slower healing behaviors (e.g., self-healing efficiency ~70% after 3600 s, Supplementary Fig. 12a). Consequently, we can conclude that the exceptionally superior self-healing characteristics of the 92-AAHA originate from the balance and synergistic effect of ion-cluster-mediated protocol and hydrogen bonding interaction.

Change: To address reviewer 1’s point, the following content has been updated in the revised manuscript.

“The fraction of P[AA][TFSI] in the AAHA copolymer is another important factor that affects the

ion cluster-mediated contribution and overall healing performance. Thus, we prepared two additional ionic conductors based on (co)polymers of 100-AAHA and 61-AAHA including different amounts of P[AA][TFSI]. The self-healing efficiency of 100-AAHA-IL with P[AA][TFSI] homopolymers was poorer than that of 92-AAHA-IL, despite its largest number of sites (*i.e.*, ammonium groups) for interacting with [EMI][TFSI] (Supplementary Fig. 12a) and lowest T_g (Supplementary Fig. 12b). On the other hand, when more PEHA units capable of forming hydrogen bonds were incorporated into AAHA, the contribution of hydrogen bonding to self-healing increased. As a result, the ionic conductor (*e.g.*, 61-AAHA-IL) became stiffer with increasing T_g (Supplementary Figs. 12b and 12c). The stiffer 61-AAHA-IL was not suitable for rapid self-healing systems, owing to the lower chain mobility (Supplementary Fig. 12a). Therefore, we can conclude that the balanced synergistic effect of the ion-cluster-mediated process and hydrogen bonding is critical to achieve a superior overall self-healing performance, leading to the 92-AAHA-IL being the optimal choice.” (line 25-28 on page 7 and line 1-10 on page 8 in the revised manuscript)

4. Can the ultra-fast self-healing ionogel be used in extreme conditions (*e.g.*, low temperature, underwater)?

We agree that this comment is very interesting and important. Therefore, we tested the self-healing behavior of the ionic conductor in extreme conditions (*e.g.*, sub-zero temperature and underwater). In both cases, the successful self-healing behavior was demonstrated with our ion-cluster-mediated self-healing system, indicating broad applicability for practical uses. The results are provided as Supplementary Video 5 and 6, and their snapshots (Supplementary Fig. 24).

Change: To address reviewer 1’s point, the following contents (video, figure, and text) have been added in the revised manuscript.

Supplementary Fig. 24 | Successful self-healing of the 92-AAHA-IL in extreme environments: **a** sub-zero temperature, and **b** underwater environment.

“Very interestingly, the 92-AAHA-IL could be successfully repaired in extreme environments such as sub-zero temperatures (Supplementary Fig. 24a and Video 5) and underwater conditions (Supplementary Fig. 24b and Video 6).” (line 19-21 on page 14 in the revised manuscript)

5. Please double check the spelling and grammar throughout the entire manuscript.

We carefully revised the entire manuscript again. This revised manuscript has also been grammatically reviewed by a native speaker.

Reviewer #2 (Remarks to the Author)

the paper "Ion-Cluster-Mediated Ultrafast Self-healable Ionoconductors for Reconfigurable Electronics" describes a material with rapid self-healing abilities (<min) at low temperature (RT). the paper is written as a step-by-step story where each points are adressed to convince the reader on the potentialities of the material ; especially, the synthesis/characterizations of material, their properties, the major role of EMI:TFSI in the self-healing behavior.

According to me the paper deserves to be published in Nature Comm. because the demonstration of superior self-healing is evidenced all along, compared to other categories of materials. Furthermore the broad applications are also addressed showing all potentialities.

here I have some suggestions/comments that authors must reply :

We appreciate the reviewer 2 for his/her recommendation of our work for publication. We believe that the original manuscript has been greatly improved by addressing the reviewer's comment.

1. EMI:TFSI appear as a "magic" tandem in so many publications. please, explain in the manuscript why use this couple of ions ; is it so specific to the low temp and rapid self-healing abilities? other ionic liquid tandem would have given similar results ? (i.e. other imidazolium, or pyridiniums or phosphoniums etc... with several families of anions of the long list of IL). Can the author provide at least an example that could demonstrate that IC-mediated self-healing is specific to the EMI:TFSI or (on the contrary) could be enlarged to broader family of IL.

Thank you for this important comment. Our criteria for the selection of [EMI][TFSI] were its low melting point ($-15\text{ }^{\circ}\text{C}$) and high conductivity (9.2 mS/cm at 25°C) (*Chem. Lett.* **8**, 922 (2000)), because the rapid healing by the ion-cluster-mediated mechanism is mainly attributed to the easy dissociation/association of ions. However, we totally agree that the effect of the other ionic liquids on the self-healing performance needs to be investigated to support our claim. Therefore, we further examined the dependence of self-healing performance on types of ionic liquids. For this purpose, as the reviewer 2 suggested, we selectively varied cations (*e.g.*, [BMI][TFSI], [P₁₄][TFSI], and [N₄₁₁₁][TFSI]) or anions (*e.g.*, [BMI][TFSI], [BMI][BF₄], and [BMI][PF₆]) of ionic liquids. Among several ionic liquid candidates, the use of [EMI][TFSI] resulted in the highest self-healing performance, irrespective of structures of cations and anions (Supplementary Fig. 18). Considering that the higher molar conductivity of ionic liquids means easier ion-ion dissociation and ion movement, the highest molar conductivity (λ) of [EMI][TFSI] is likely to be a key factor for the fastest self-healing behavior. Indeed, when plotted the molar conductivity of the used ionic liquid

and the self-healing efficiency of the corresponding ionic conductor, their intimate correlation was found (Supplementary Fig. 18).

Change: To address the reviewer 2's comment, the following contents have been added in the revised manuscript and Supplementary Information.

Supplementary Fig. 18 | Time-dependent self-healing efficiency of 92-AAHA-IL with selectively varied **a** cations and **b** anions of the ionic liquid. **c** Plots of molar conductivity (λ) and healing efficiency of ionic conductors at a repair time of 60 s. The λ values were extracted from ref. S24. The self-healing processes of all ionic conductors were performed at 25 °C. Considering that a higher λ of an ionic liquid denotes an easier ion-ion dissociation and ion movement, the superior performance of the [EMI][TFSI]-containing 92-AAHA-IL can be rationalized in terms of the highest λ of the ionic liquid additive. All error bars indicate standard deviation.

“The importance of a facile ion dissociation/association was further supported by the results of experiments performed with various ionic liquid additives (Supplementary Fig. 18), in which we varied the cations and anions of the ionic liquid. However, the fastest recovery was still observed with [EMI][TFSI], which can be explained by its highest molar conductivity related to ion-ion dissociation and ion movement.” (line 20-24 on page 10 in the revised manuscript)

2. Figure 1a : the star point "This Work" is really out of any other published studies and seems to prove the breakthrough in the area of self-healing materials. So authors define the current work as a new point of an existing family in the graph or they claim this is a new category of self-healable materials.

We believe that ion-cluster(IC)-mediated ionconductors can become a new category of self-healing materials due to their different healing mechanism and superior performance. Therefore, we defined our system as an ‘IC-mediated Ionconductor’ in Fig. 1a.

Change: To address reviewer 2’s comment, the following Fig 1a has been updated in the revised manuscript.

Fig. 1 | a Trade-off correlation between mechanical robustness and self-healing speed present in conventional self-healing systems.

3. Application: all examples of application in the manuscript give totally impressive results and videos are impressive too. Now if I look for my own concern, one application, supercapacitor, could have been explored: typically how is evolving the specific power of supercapacitor after 10, 20, 50 self-healing steps. This is crucial and could be also easily addressed here. (Evolution of specific power would be mainly resulting of ionic behavior during self-healing steps)

We agree with the reviewer 2’s comment that our self-healing ionconductor can be also applied to supercapacitor. Therefore, we fabricated electrical double layer capacitors (EDLCs) using the 92-AAHA-IL as an electrolyte (Supplementary Fig. 26a) and investigated the change in their energy storage characteristics as a function of the number of self-healing. When we recorded the

cyclic voltammogram (CV) of the EDLC, a rectangular-shaped curve was shown similar to conventional EDLCs (Supplementary Fig. 26b). An ideal galvanostatic charge-discharge (GCD) curve was also measured from the device (Supplementary Fig. 26c). These observations imply that the 92-AAHA-IL successfully serves as an electrolyte for EDLCs. No noticeable change in the shape of both CV and GCD curves was displayed even after 50-times cut-and-healing cycles (Supplementary Fig. 26b and 26c), leading to high and constant capacitance retention (C/C_0) irrespective of the number of healings (Supplementary Fig. 26d). Overall, the results demonstrated high applicability of the 92-AAHA-IL in self-healable energy storage devices.

Change: To update reviewer 2's comment, the following contents have been provided in the revised manuscript and Supplementary Information.

Supplementary Fig. 26 | **a** Schematic illustration of self-healable ionoconductor-based electrical double layer capacitor (EDLC). The inset corresponds to a photograph of the actual device. **b** Cyclic voltammogram (CV) curves at a scan rate of 25 mV s^{-1} during self-healing cycles. The areal capacitance (C) was calculated using the following equation:^{S28} $C = \frac{1}{\Delta V(dV/dt)} \int I dV$, where dV/dt and $\int I dV$ are the voltage scan rate and the numerical integration of the current density during the half-cycle potential window (ΔV), respectively. **c** Galvanostatic charge-discharge (GCD) profiles at a current density of $0.1 \text{ mA}\cdot\text{cm}^{-2}$. The GCD capacitance was extracted using the equation:^{S29} $C = I\Delta t/S\Delta V$, where I , Δt , S , and ΔV represent the discharging current, discharging time, electrode area, and potential window excluding the IR voltage drop, respectively. **d** Variation in capacitance retention (C/C_0) as a function of the number of healings. Irrespective of self-healing processes, high C/C_0 was maintained, indicating practical applicability of the 92-AAHA-IL in self-healable energy storage devices.

“Moreover, the high thermal stability (Supplementary Fig. 25) and successful application of 92-AAHA-IL in self-healable energy storage devices (*e.g.*, supercapacitors, Supplementary Fig. 26) expanded its practical application range.” (*line 21-24 on page 14 in the revised manuscript*)

“Fabrication and characterization of self-healable supercapacitors. To assess the applicability in energy storage systems such as supercapacitors, the 92-AAHA-IL film (thickness: 2 mm, width: 10 mm, and length: 10 mm) was employed as a self-healable electrolyte for an electrical double layer capacitor (EDLC). To enlarge the electrode surface, ITO particles were additionally spin-coated (1200 rpm, 60 s) onto the ITO-coated glass and vacuum-dried at 60 °C. The fabrication of the supercapacitor was completed by sandwiching 92-AAHA-IL between two ITO particle-decorated electrodes. The performance of the EDLC was characterized using cyclic voltammetry (CV, Wave Driver 10, Pine Instrument) and galvanostatic charge-discharge (GCD, WBCS30000L, WonATech) measurements.” (*line 15-23 on page 18 in the revised manuscript*)

REVIEWERS' COMMENTS

Reviewer #1 (Remarks to the Author):

The authors have addressed the issues that reviewers proposed. And the manuscript has qualified for its publication.

Reviewer #2 (Remarks to the Author):

After careful read of the replies-to-comments document, I estimate that authors have totally answered the points I raised.

Furthermore, I am impressed the new data obtained on supercapacitors behavior after some tens of cutting/self-healings steps.

I have no more comments and I strongly recommend the paper to be published as is.

Response to Review and Reference to Changes for NCOMMS-22-08900A

Reviewer's comments

Reviewer #1 (Remarks to the Author):

The authors have addressed the issues that reviewers proposed. And the manuscript has qualified for its publication.

We thank the reviewer 1 for supporting the publication of this work as is.

Reviewer #2 (Remarks to the Author):

After carefull read of the replies-to-comments document, I estimate that authors have totally answered the points I raised. Furthermore, I am impressed the new data obtained on supercapacitors behavior after some tens of cutting/self-healings steps. I have no more comments and I strongly recommand the paper to be published as is.

We appreciate the reviewer 2 for his/her strong recommendation of this work for publication as is.